# Effect of Boron on the Microstructure, Superplastic Behavior, and Mechanical Properties of Ti-4Al-3Mo-1V Alloy

**DOI:** 10.3390/ma16103714

**Published:** 2023-05-13

**Authors:** Maria N. Postnikova, Anton D. Kotov, Andrey I. Bazlov, Ahmed O. Mosleh, Svetlana V. Medvedeva, Anastasia V. Mikhaylovskaya

**Affiliations:** 1Department of Physical Metallurgy of Non-Ferrous Metals, National University of Sciences and Technology “MISIS”, 4 Leninskiy Ave. 4, 119049 Moscow, Russia; sitkina.m@misis.ru (M.N.P.); bazlov@misis.ru (A.I.B.); medvedeva71@list.ru (S.V.M.); mihaylovskaya@misis.ru (A.V.M.); 2Department of Mechanical Engineering, Faculty of Engineering at Shoubra, Benha University, Cairo 11629, Egypt; ahmed.omar@feng.bu.edu.eg

**Keywords:** titanium alloys, superplasticity, TiB, ultrafine-grained structure, thermomechanical treatment, metal matrix composites

## Abstract

The decrease of superplastic forming temperature and improvement of post-forming mechanical properties are important issues for titanium-based alloys. Ultrafine-grained and homogeneous microstructure are required to improve both processing and mechanical properties. This study focuses on the influence of 0.01–2 wt.% B (boron) on the microstructure and properties of Ti-4Al-3Mo-1V (wt.%) alloys. The microstructure evolution, superplasticity, and room temperature mechanical properties of boron-free and boron-modified alloys were investigated using light optical microscopy, scanning electron microscopy, electron backscatter diffraction, X-ray diffraction analysis, and uniaxial tensile tests. A trace addition of 0.01 to 0.1 wt.% B significantly refined prior *β*-grains and improved superplasticity. Alloys with minor B and B-free alloy exhibited similar superplastic elongations of 400–1000% in a temperature range of 700–875 °C and strain rate sensitivity coefficient *m* of 0.4–0.5. Along with this, a trace boron addition provided a stable flow and effectively reduced flow stress values, especially at low temperatures, that was explained by the acceleration of the recrystallization and globularization of the microstructure at the initial stage of superplastic deformation. Recrystallization-induced decrease in yield strength from 770 MPa to 680 MPa was observed with an increase in boron content from 0 to 0.1%. Post-forming heat treatment, including quenching and ageing, increased strength characteristics of the alloys with 0.01 and 0.1% boron by 90–140 MPa and insignificantly decreased ductility. Alloys with 1–2% B exhibited an opposite behavior. For the high-boron alloys, the refinement effect of the prior *β*-grains was not detected. A high fraction of borides of ~5–11% deteriorated the superplastic properties and drastically decreased ductility at room temperature. The alloy with 2% B demonstrated non-superplastic behavior and low level of strength properties; meanwhile, the alloy with 1% B exhibited superplasticity at 875 °C with elongation of ~500%, post-forming yield strength of 830 MPa, and ultimate tensile strength of 1020 MPa at room temperature. The differences between minor boron and high boron influence on the grain structure and properties were discussed and the mechanisms of the boron influence were suggested.

## 1. Introduction

Titanium alloys are widely used in the aircraft, transport, medicine, and chemical industries due to their high specific strength and corrosion resistance. However, obtaining parts from titanium alloys with traditional methods at low temperatures is a difficult and energy-consuming process due to their high strength, low elasticity modulus, and high sensitivity to processing parameters [1]. Superplastic forming (SPF) is an alternative to traditional forming methods which produces parts of complex geometry in one forming operation at a low gas pressure and with high dimensional accuracy [2,3]. The superplastic behavior of titanium alloys depends on the grain size and shape and *α*/*β* phase ratio [4,5]. Control of the grain structure and its evolution are required to ensure excellent superplasticity. Grain size control is an important issue from the step of solidification to thermomechanical processing and superplastic deformation.

Alloying with 0.05–0.2 B (boron, wt.%) improves the processing and mechanical properties of titanium-based alloys [6,7,8,9]. Boron is low-soluble (up to 0.02 wt.% [10,11]) in titanium and it forms fine particles of the TiB phase [12,13,14]. The TiB particles solidify at eutectic transformation and exhibit the elongated shape of “whiskers” [15,16]. Trace addition of boron in titanium and titanium-based alloys provides a strong grain refinement effect during solidification [8,9,17]. The grain refinement mechanism is related to the TiB phase solidification [18,19,20,21,22]. According to the Ti-B phase diagram, hypoeutectic *β*-Ti grains solidify first and the eutectic-originated TiB phase could not stimulate *β*-Ti phase nucleation in the liquid phase with B content below ~2% [22]. Larson presumes [23] that the borides can act as inoculants, i.e., the ingot re-melts several times and eutectic-originated borides do not dissolve during the re-melting process. The most acceptable refinement mechanism is related to a refinement of dendrite arm spacing due to B atomic segregations at the front of solidification and constitutional supercooling effect [24]. The Zener pinning effect of TiB particles in a solid state is also possible [25]. A trace amount of 0.02% B refines grains for Ti–6Al–4V alloy [26,27] and the effect intensifies up to 0.1% B [24]. The grain size for alloys with 0.1% B is similar for 0.4% B; therefore, the 0.1% B is considered the most effective [24].

For wrought alloys, boron provides globularization of the microstructure during heat and thermomechanical treatments. The proposed mechanism is stimulation of the *α* phase nucleation near TiB particles [28,29]. The authors of [16] observe boron atomic segregations at the *α/β* interfaces and reasonably assume the ‘boron solute pinning mechanism’ for *α*-phase colonies.

The TiB phase has an equivalent density to titanium but five times greater strength [30]. Notably, the TiB phase is stable over a wide temperature range [30]. These particles are efficient reinforcers at a considerable volume fraction [31]. An increase in boron content provides the natural composite structure of the Ti-based alloys reinforcements with in situ precipitated TiB whiskers. High-boron alloys have significant practical importance and their microstructure and properties have been studied but scarcely [9,30,32].

Thus, minor boron refines the grain structure in the as-cast state as well as after thermo-mechanical processing and increases the homogeneity of the microstructure in the ingots and thermomechanically treated products [7,9,28,32,33,34]. The effectiveness of a trace amount of boron for grain refinement of unalloyed Ti and several Ti-based alloys was confirmed. The B effect depends on the alloy composition, residual elements [23], and treatment parameters, and further investigations for the alloys with different boron content and different chemical compositions are required to convince the industry of the effectiveness of B alloying.

A conventional Ti-4Al-3Mo-1V alloy is widely applied due to its high strength, excellent corrosion resistance, and good creep resistance. The alloy exhibits superplasticity in a temperature range of 825–875 °C similar to Ti-6Al-4V [35,36]. In contrast to the widely used Ti-6Al-4V alloy, the deformation behavior of Ti-4Al-3Mo-1V alloy is characterized by strain softening, owing to dynamic recrystallization, that complicates superplastic forming [37]. Due to grain refinement, a trace addition of 0.1% B improves superplastic properties and increases the superplastic formability of the Ti-6Al-4V alloy [28]. The complex alloying with B and Fe improves superplasticity and decreases superplastic deformation temperature for Ti-4Al-3Mo-1V alloy, but the effect of B for complexly alloyed materials is not clear [38]. This work studies the influence of boron in a range of 0.01–2% on the microstructure, superplasticity, and mechanical properties at room temperature of Ti-4Al-3Mo-1V.

## 2. Materials and Methods

### 2.1. Material Processing

The boron-free reference Ti-Al-Mo-V alloy and five alloys with boron additions in a range of 0.01–2 wt.% were prepared (Table 1). The ingots were processed by argon arc melting in a vacuum laboratory furnace Arc Melter ARC200 (ARCAST Company, Oxford, MS, USA). The concentrations of Al, Mo, and V alloying elements in the alloys were similar. Titanium (>99.9 wt.%), aluminum (>99.99 wt.%), vanadium (>99.9 wt.%), boron (>99.99 wt.%), and the Ti–50 wt.% Mo master alloy were used to prepare the alloys. The ~100 g ingots were re-melted five times to ensure a homogeneous alloy composition and cast into a copper water-cooled mold with an internal size of 50 × 40 × 10 mm^3^. The ingots were subjected to homogenization annealing in a furnace with a vacuum atmosphere at 800 °C for 1 h and subsequently heated to the *β*-field and water-quenched.

Quenched ingots of the 0B, 0.01B, 0.05B, and 0.1B alloys were hot-rolled with a total reduction of 90% at a temperature of 750 ± 10 °C. A low ductility did not allow us to roll the high-boron alloys at 750 °C. Ingots of the 1B and 2B alloys (with 1–2% B) were successfully rolled at a higher temperature of 900 ± 10 °C. The sheets were treated in Kroll’s reagent (92% H_2_O + 3% HF + 5% HNO_3_) for 20–30 min to remove the *α*-phase layer from the surface. The final thickness of the hot-rolled sheets was 1.0 ± 0.1 mm.

### 2.2. Microstructural and Phase Composition Analyses

The as-cast grain structure of the alloys was analyzed using a Neophot-30 optical (light) microscope (LM) with polarized light. To analyze the phase composition and grain size of the thermomechanical treated alloys, a Vega 3-LMH scanning electron microscope (SEM) (Tescan Brno s.r.o., Kohoutovice, Czech Republic) equipped with the EDS X-MAX 80 energy dispersive spectrometer (Oxford Instruments plc, Abingdon, UK) was used. The microstructural parameters of the *α*, *β*, and TiB particles were studied after hot rolling and after annealing in a vacuum furnace at 700 °C, 775 °C or 875 °C for 30 min and subsequent water quenching. The mean size and volume fraction of grains and TiB particles were determined by random linear secants method by averaging more than 200 measurements. The error bars were calculated using standard deviation of a mean value and a confidence probability of 0.95. The grain and sub-grain structure of the thermomechanical processed alloys were also analyzed with backscattered electron diffraction (EBSD) technique using an HKL NordlysMax EBSD detector (Oxford Instruments plc, Abingdon, UK). The scanning area was 50 × 50 μm^2^ and scanning step was 0.15 μm. The EBSD data, i.e., grain and subgrain sizes (equivalent diameter), were analyzed using HKL CHANNEL 5 software (version 5.11.20405.0, Oxford Instruments plc, Abingdon, UK). Grain boundaries with a misorientation angle (*θ*) of 2° < *θ* < 15° were classified as low-angle grain boundaries (LAGBs), and grain boundaries with *θ* ≥ 15° were classified as high-angle grain boundaries (HAGBs).

Samples for microstructural analysis were prepared by mechanical grinding and polishing on a CHEM MD cloth in a solution (silicon oxide suspension 50 mL, H_2_O_2_ (30%) 10 mL, and Kroll reagent 5 mL) on a Struers LaboPol-5 (Struers APS, Ballerup, Denmark). To identify grain structure in an as-cast state, the samples were etched with Weck’s reagent (H_2_O 100 mL, ethanol 50 mL, and ammonium biflouride NH4FHF 2 g).

XRD analysis was performed using a D8 ADVANCE X-ray diffractometer (Bruker, Billerica, MA, USA) in a Cu-Kα radiation. To calculate the equilibrium *β*-transus temperature for the studied alloys a Thermo-Calc (Thermo-Calc Software, version, 5.0.4.75, Stockholm, Sweden) software and a TTTI3 database were used.

### 2.3. Tensile Tests

The superplastic deformation behavior was characterized by uniaxial tension tests at a constant rate and a step-by-step decrease in the strain rate at temperatures of 875 °C, 775 °C and 700 °C on a Walter Bai LFM-100 (Walter + Bai AG, Löhningen, Switzerland) universal testing machine in a furnace with an argon atmosphere. Samples with the gauge section size dimensions of 14 × 6 × 1 mm^3^ were cut parallel to the rolling direction and annealed for 30 min at the superplastic deformation temperatures before testing.

The mechanical properties of the alloys at room temperature were determined on a Zwick Z250 testing machine (Zwick Roell Group, Ulm, Germany) using samples with a gauge section size of 6 × 0.8 × 25 mm^3^. Before cutting the samples, the sheets were pre-strained for 100% at a temperature of 875 °C with a constant strain rate of 1 × 10^−3^ s^−1^. Then, two types of treatment were processed: (1) air-cooling from the deformation temperature, and (2) water-quenching from the deformation temperature and further ageing at 480 °C for 16 h. Before testing, the samples were treated in Kroll’s reagent for ~20 min for *α*-case removal. Three samples per point were tested.

## 3. Results

### 3.1. Phase Composition Analysis

The boron content for the studied alloys was chosen in a hypoeutectic concentration range for (0.01–1%) B and at the eutectic point for 2% B according to Thermo Calc simulated polythermal cross section of the multicomponent Ti-Al-Mo-V-B phase diagram (for Ti-4Al-3Mo-1V-(0–4%) B nominal composition) (Figure 1a). XRD phase analysis for the alloys with trace boron addition of below 0.1% and boron-free alloy identified the existence of *α* and *β* phases. Due to a high boron content in the 1B and 2B alloys, the clear peaks of the TiB phase were revealed. The peaks of low intensity attributed to titanium boride were found in the alloy with 0.1% B, and this phase was not observed in the alloys with 0.01–0.05% B.

### 3.2. Analysis of The Microstructure after Solidification, Thermomechanical Treatment, and Post-Deformation Annealing

The grain structure of the alloys in the as-cast state is shown in Figure 2. The addition of a trace amount of 0.01% B resulted in a decrease in the size of the primary *β* phase grains from 700 ± 70 μm to 490 ± 40 μm (Table 2). An increase of the boron content to 0.1% led to a significant decrease in grain size to 210 ± 20 μm (Table 2). Surprisingly, for the alloys with a high boron content of 1–2%, the average grain size was 710 ± 50 μm, i.e., the value was similar to the boron–free alloy.

SEM analysis of the alloys identified the matrix of the transformed prior *β*-grains and dark particles for all B-containing alloys after solidification (Figure 3). The EDS analysis revealed an increased boron content for the dark particles that indicated the TiB phase (Figure 4).

The volume fraction of the TiB phase increased from 0.1 to 1.1% with an increase in boron content from 0.01 to 0.1%. Rarely distributed fine TiB particles were found in the 0.01B alloy, even at 0.01% B (Figure 3b). The individual TiB particles and their agglomerations were observed in the alloys with 0.05–0.1% B. The volume fraction of TiB particles was 5.0 and 10.8% for high-boron alloys with 1 and 2% B, respectively (Table 2). The TiB particles were uniformly distributed in the matrix of the alloy, with 2% B corresponding to eutectic composition (Figure 3f). The borides’ morphology varied from compact polyhedral inclusions to elongated whiskers. For the 1B and 2B alloys, the TiB particles after solidification exhibited an elongated shape with a longitudinal size up to 9 μm, and finer and more compact-shaped borides were observed for the alloys with a low boron content of 0.01–0.1%. An average particle size increased from 0.7 to 2.5 µm with an increase in boron content from 0.01 to 0.1–1% and insignificantly grown to 2.9 µm for 2% B.

The microstructure of the alloys studied after thermomechanical treatment was characterized by elongated grains of the *α* and *β* phases. The mean transverse sizes of both phases were similar for the studied alloys, the *α*-phase grains were in a range of 0.9–1.2 µm and the *β*-phase grains were in range of 0.5–0.8 µm (Figure 5, Table 3). The main difference between the microstructures of the rolled alloys was in the size and volume fraction of TiB particles (Table 3). Particles of the TiB phase in alloys with 0.1–1% B exhibited an elongated shape with a length (*L_∥_*) of 2.1–3.1 μm and width (*L_⊥_*) of 0.6–0.7 μm and a low aspect ratio (*L_⊥_/L_∥_*) of 0.2–0.3. In contrast, the morphology of TiB particles in the eutectic 2B alloy was characterized by a near-spherical shape with an aspect ratio (*L_⊥_/L_∥_*) equal to 1 and an average size of 1.0 μm.

Annealing at 875 °C for 30 min, which led to the microstructure before the onset of superplastic deformation, did not influence on the morphology and fraction of borides. Meanwhile, the microstructure of boron-free and boron-bearing alloys were different (Figure 6). The microstructure of the B-free alloy was inhomogeneous. A high non-recrystallized fraction with elongated grains of both *α* and *β* phases was observed (Figure 6a–c). The alloy with 0.1% B was characterized by a globular equiaxed microstructure (Figure 6d–f). EBSD analysis revealed a significant difference in the grain/subgrain structures of 0B and 0.1B alloys (Figure 7). The fraction of low-angle grain boundaries (LAGBs) was 40% (Figure 7a,b), the mean kernel average misorientation (KAM) angle was 0.82° (Figure 7c,d) and a substructured volume (Figure 7e) was dominant for B-free alloy. The smaller LAGBs fraction of 22% (Figure 7f,g) and smaller mean KAM angle of 0.46° (Figure 7h,i) were revealed, and recrystallized volume dominated for 0.1B alloy (Figure 7j). The mean grain/subgrain sizes for the *α* phase with HCP structure (including *α* and transformed *β*) were 1.0 µm/0.6 µm for 0B and 1.7 µm/1.1 µm for 0.1B alloy.

### 3.3. Superplastic Deformation

Alloys with 0.01–0.1% B, similar to the reference B-free alloy, exhibited superplasticity at 875 °C with a strain rate sensitivity coefficient *m* ~0.5 in a strain rate range of 5 × 10^−4^ to 5 × 10^−3^ s^−1^. Elongation to failure of ~1000% was received at a constant strain rate of 1 × 10^−3^ s^−1^ (Figure 8 and Figure 9). At the lower deformation temperature of 775 °C, the maximum *m* shifted to the lower strain rates of (2–4) × 10^−4^ s^−1,^ and elongation to failure decreased to 500–600%. At a temperature of 700 °C, the deformation was characterized by a lower coefficient of *m* ~0.4 and elongations of 400–500%. Due to the trace boron addition of 0.01–0.1%, the elongation to failure changed insignificantly at 875 °C and slightly increased at 700–775 °C. Moreover, alloys with minor boron addition exhibited significantly smaller flow stresses at the initial stage of deformation than that of the B-free alloy. The differences in the deformation behavior of the alloys were most pronounced at a temperature of 775 °C. At the initial stage (in a range of ε = 0.1–0.7), the deformation of B-free alloy was accompanied by strain softening but a steady flow was observed for the alloys with 0.01–0.1% B (Figure 8e).

Alloys with 1 and 2% B were characterized by lower values of the strain rate sensitivity coefficient *m* than alloys with 0.01–0.1% B at 875 °C; the maximum *m* value was 0.42 at 1 × 10^−3^ s^−1^ for the 1B alloy and 0.35 at 2 × 10^−4^ s^−1^ for the 2B alloy (Figure 8c). The 1B alloy demonstrated superplastic behavior at the deformation with the strain rate corresponding to the maximum *m*, with elongation to failure of ~500%. The 2B alloy demonstrated non-superplastic behavior even at a low strain rate of 2 × 10^−4^ s^−1^ at elevated temperature of 875 °C; the flow stress for the 2B alloy was higher than for the 1B alloy and the elongation was twice smaller (250%).

### 3.4. Mechanical Properties at Room Temperature

The mechanical properties at room temperature of the alloys studied (yield strength *YS*, ultimate tensile strength *UTS*, and elongation to fracture *δ*) were determined after deformation at a temperature of 875 °C with a constant strain rate of 1 × 10^−3^ s^−1^ (2 × 10^−4^ s^−1^ for the 2B alloy) for a strain of 0.69 (100% of engineering strain) (Table 4). Increasing the boron content decreased the yield strength from 770 MPa for B-free alloy to 680 MPa for 0.1%. The boron in a range of 0.01 to 0.1% insignificantly influenced the UTS and elongation at fracture, which were 840–870 MPa and 6–7%, respectively. Increasing boron to 1% B improved both strength characteristics, with YS increasing to 830 MPa and UTS to 1020 MPa, but drastically dropped ductility to a critically low value of elongation to fracture of ~1%. Alloy with 2% B demonstrated a brittle fracture and resulted in smaller strength properties. Quenching and aging of the alloys with a low boron content of 0.01 and 0.1% increased yield strength by 100–130 MPa and tensile strength by 90–140 MPa. For the 0.01B alloy, elongation at fracture was the same after both air cooling and heat treatment, but for the 0.1B alloy, ductility slightly reduced after heat treatment from 7 to 5%.

## 4. Discussion

### 4.1. Boron Influence on The Prior β-Grain Size after Solidification

A trace boron addition, up to 0.1%, provided an effective refinement of the prior *β* grains. The mean grain size decreased more than three times (Table 2). The grain refinement effect of 0.02–0.2% boron is well known for unalloyed Ti [38], several Ti-based alloys [16,24], and Ti-Ni [39,40] and Ti-Al [41,42,43] intermetallic alloys. The difference in the grain size between 0.1 and 0.4 wt.% B was insignificant. The boron at the same content of ~0.1% was effective for the grain refinement in the studied Ti-Al-Mo-V alloy. The mechanisms of the B effect for as-cast grain refinement are attributed to the formation of boride particles, which can inhibit grain growth in a *β* phase field [16,44] or B segregations at the front of solidification due to its extra-low solubility in Ti and distribution coefficient <1 [44,45]. In addition, increasing boron content leads to a decrease in the liquidus temperature of the alloys and a narrowing of the solidification range (Figure 1). This effect should increase the crystallization rate and stimulate nucleation kinetics. A similar grain refinement effect with increasing alloying element content was observed for Al-Mg alloys [46] and Ni-Cu alloys [47]. For the studied hypoeutectic alloys, it was demonstrated that the grain refinement effect disappeared at high boron content of 1–2% B. The mechanism of the phenomenon is unclear. With an increase in boron content, the fraction of TiB particles significantly increased (Table 2), considering the same mean sizes of the boride particles for 0.1% B and 1% B of 2.5 µm. The Zener pinning effect increases with the increase in particle fractions and with decreased particle size [25]. Thus, during cooling in a *β*-phase field, Zener pinning force should be stronger and finer grains are expected. Meanwhile, grains were coarser at a high borides fraction. Thus, the refinement effect could not be explained by the Zener pinning mechanism. Similarly, the effect of segregation of boron at the periphery of *β* dendrites during their growth in a liquid phase should be stronger with higher B content. The microstructure and boride morphology suggested the same origin of the TiB phase precipitation via eutectic transformation for 0.1% B and 1–2% B. A similar B concentration in eutectic point was observed for Ti-B alloys [30] and slightly smaller ~1.6% B for Ti-6Al-4V-B alloys [32]. The solidification range decreases with the increase of boron from 0.1 to 2% and became narrow for the eutectic concentration of ~2% B. Thus, the nature of the degradation of grain refinement effect for high-boron hypoeutectic alloys is unclear and requires explanations and further investigations. The observed phenomenon also suggests that further clarification of the grain refinement mechanism for prior *β*-grains via trace boron additions is required. The possible explanation is as follows. First, considering the widely acceptable constitutional supercooling effect, trace boron addition refines the *β*-phase dendrites, and the change of the crystal growth type from dendritic to eutectic bicrystal with colonies of two phases eliminates this effect. At a high boron content and a high fraction of *β* phase of eutectic origin, the effect disappeared due to extra-rapid simultaneous growth of TiB and *β* colonies. At minor concentrations, boron atoms in the liquid may stimulate the nucleation of the *β* phase due to a decrease of the liquid–solid interphase energy or a decrease in the critical size of nuclei due to B atomic segregations [16,38]. A segregation effect is the most pronounced at the minor content of alloying element or impurity and the distribution of the elements became more homogeneous at its high content [16]. Second, the widely accepted theory for grain refinement during solidification for various alloys is still the inoculation effect, e.g., Ti with B in Al [48], Zr in Mg, or Al [49]. This mechanism seems impossible for hypoeutectic Ti-B alloys when *β*-phase grains are solidified before the eutectic-originated TiB phase [50]. Meanwhile, the ingots of the studied alloys were re-melted five times, and for low-boron alloys with a high liquidus temperature, high-temperature borides may incompletely dissolve during the re-melting process and provide a heterogeneous nucleation effect. This effect has been noted in [23] as Larson’s theory. At a high fraction of Ti+TiB eutectic, TiB colonies rapidly dissolute due to a short-range diffusion and narrow solidification range, smaller liquidus temperature with a higher overheating degree. Third, for low-boron alloys, the precipitation of primary B-induced metastable phases due to non-equilibria solidification or other unconsidered phases with interstitial impurities in a liquid phase may provide heterogeneous nucleation of the *β*-phase during solidification of the studied alloy. Even a small fraction of such phases is enough to stimulate inoculation. The formation of the thermodynamically stable borides simplified the high-boron 1B and 2B alloys and no inoculation effect was observed.

### 4.2. Influence of Trace Addition of 0.01–0.1% B on The Microstructure and Superplasticity

Trace boron alloying also significantly changed the as-annealed microstructure of the thermomechanical processed alloys. The B-free alloy demonstrated an inhomogeneous structure with a mixture of the elongated and equiaxed grains and a high fraction of low-mobile LAGBs. Alloying with 0.1% B provided a globular microstructure in the alloy. A similar effect was found due to alloying of Ti-Al-Mo-V alloy with both Fe and B [38]. Thus, boron plays a key role in the acceleration of the recrystallization and further globularization of the grains. These results are consistent with [51], indicating that TiB particles accelerate the microstructure globularization due to particle stimulation nucleation of the α phase [31]. The B atomic segregations at the *α/β* interfaces, which were recently found in [16], may also facilitate recrystallization and globularization of the microstructure during annealing. Trace B addition twice decreased a fraction of low-mobile LAGBs and increased a fraction of high-mobile HAGBs. A decrease of the flow stress at the initial stage of superplastic deformation due to B trace addition was the result of recrystallized ultrafine-grained structure formation before the onset of deformation. As the result, the studied alloys with 0.01–0.1% B exhibited smaller flow stresses than the studied reference 0B alloy and conventional Ti-4Al-3Mo-1V alloy [35,37]. The formation of ultrafine equiaxed grains with high-angle grain or interphase boundaries facilitated grain boundary sliding [52]. The boron influence on the deformation behavior was significant at a low deformation temperature of 775 °C (Figure 8d,e). Thus, alloying with trace boron is an effective strategy to improve the superplasticity of Ti-based alloys at low temperatures, which is of high practical importance for the superplastic forming process. Along with this, the trace B addition decreased the post-forming yield strength of the alloys owing to a higher recrystallized fraction and larger mean grain size than B-free alloy.

### 4.3. Influence of 1–2 wt.% B on The Superplastic and Mechanical Properties

Due to a high fraction of borides, an increase in B content to 1% increased room temperature strength but weakens superplastic properties and decreased ductility at both elevated and room temperatures. The same influence of boron on the room temperature properties was observed for Ti-6Al-4V alloy [53]. The authors of [15,54] observed an increased strength but decreased ductility and fracture toughness for the alloys with 0.4–2% B. To increase ductility and toughness, the morphology of borides should be changed from whiskers to spherical shapes. Notably, thermo-mechanical treatment of the alloy with 2% B provided refinement of TiB particles during high-temperature deformation and contributed to the formation of more compact particles. Particle refinement may be the result of both mechanical breaking [55,56] and fragmentation and spheroidization processes during hot deformation at an elevated temperature of 900 °C. The same processes occur during the spheroidization of cementite in steel [57,58] or intermetallic particles in aluminum-based alloys [59,60]. Despite their size of about 1 µm, the TiB particles induced a drop in ductility and led to an embrittlement effect in 2% B alloy. Meanwhile, the alloy with 1% B demonstrated an acceptable combination of high-temperature superplasticity at 875 °C and a high room-temperature tensile strength. To avoid embrittlement, the high boron alloys of sheet processing technologies should be focused on the morphology of borides.

## 5. Conclusions

The influence of 0.01–2 wt.% B on the microstructure, superplasticity, and mechanical properties at room temperature of the Ti-4%Al-3%Mo-1%V alloy was studied. The boron effect on the microstructure and properties of the alloys was strongly different for the ranges of 0.01–0.1% B and 1–2% B. The main conclusions are summarized as follows.

The TiB particles of eutectic origin were formed after solidification in the studied B range. In as-cast alloys, the mean size of borides increased from 0.7 to 2.5 µm with an increase in B from 0.01 to 0.1% and insignificantly changed in a range of 0.1–2% B. The volume fraction of the TiB phase increased from ~0.1% at 0.01% B to ~11% at 2% B, which agreed with the Thermo Calc simulation. Trace boron addition significantly reduced the mean size of prior *β* grains from ~700 µm for the B-free alloy to ~210 µm for the alloy with 0.1% B. Grain refinement was not shown and the mean grain size was ~670–750 µm for the alloys with 1 and 2% B.

After thermomechanical treatment with a final hot rolling, the alloys exhibited similar elongated grains of a mean size in a range of 0.9–1.2 µm for the *α* phase and 0.5–0.8 µm for the *β* phase. For the B-free and minor B alloys with TiB volume fraction below ~1.1%, the strain rate sensitivity coefficient *m* was ~0.5, and the elongation to failure was ~500–1000% at temperatures of 775–875 °C and a strain rate of 1 × 10^−3^ s^−1^, but B influenced deformation behavior. Due to the facilitation of recrystallization and globularization effects, the 0.01–0.1% B decreased flow stress values at the initial stage of superplastic deformation and provided a stable flow. The effect was significant at the low deformation temperature of 775 °C. Along with this, owing to the stimulation recrystallization effect, a trace B addition decreases post-forming strength properties and insignificantly influenced ductility at room temperature; yield strength decreased from 770 MPa for the B-free alloy to 733 MPa for 0.01% B, and to 680 MPa for 0.1% B. Quenching and aging increased the strength properties by 90–140 MPa and slightly decreased ductility for 0.1% B. The modifying of Ti-Al-Mo-V alloy with 0.01% B provided both uniform and stable flow during superplastic deformation and good mechanical properties at room temperature.

The alloy with 1% B and ~5.2% of TiB whiskers of a mean thickness ~0.7 µm and a mean length of ~3 µm exhibited superplasticity at elevated deformation temperature of 875 °C with *m* of 0.5, elongation to failure of ~500%, room temperature yield strength of 830 MPa, and ultimate strength of 1020 MPa, but low elongation at fracture of 1.1%. An increased boron content with the eutectic concentration of 2 wt.% increased the TiB particles fraction to ~11% and provided the particles with a mean size of ~1 μm after thermomechanical processing. The high-boron eutectic alloy demonstrated a non-superplastic behavior at elevated temperatures and a brittle fracture at room temperature, with a strong decrease in the ultimate tensile strength to 780 MPa.

## Figures and Tables

**Figure 1 materials-16-03714-f001:**
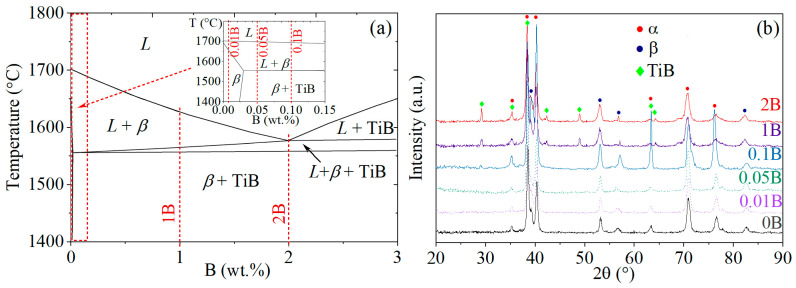
(**a**) Polythermal section of the Ti–4Al–3Mo–1V-B diagram (Thermo-Calc) and (**b**) XRD patterns for the alloys studied.

**Figure 2 materials-16-03714-f002:**
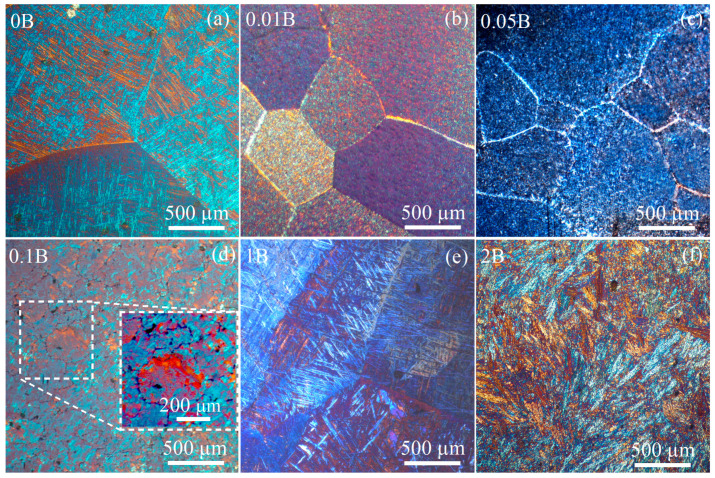
As-cast microstructure of the (**a**) 0B, (**b**) 0.01B, (**c**) 0.05B, (**d**) 0.1B, (**e**) 1B, and (**f**) 2B alloys.

**Figure 3 materials-16-03714-f003:**
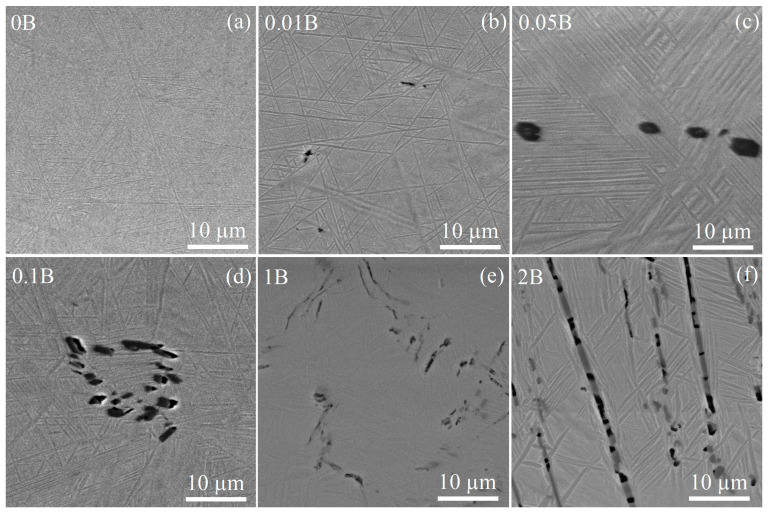
As-cast SEM images of the (**a**) 0B, (**b**) 0.01B, (**c**) 0.05B, (**d**) 0.1B, (**e**) 1B, and (**f**) 2B alloys.

**Figure 4 materials-16-03714-f004:**
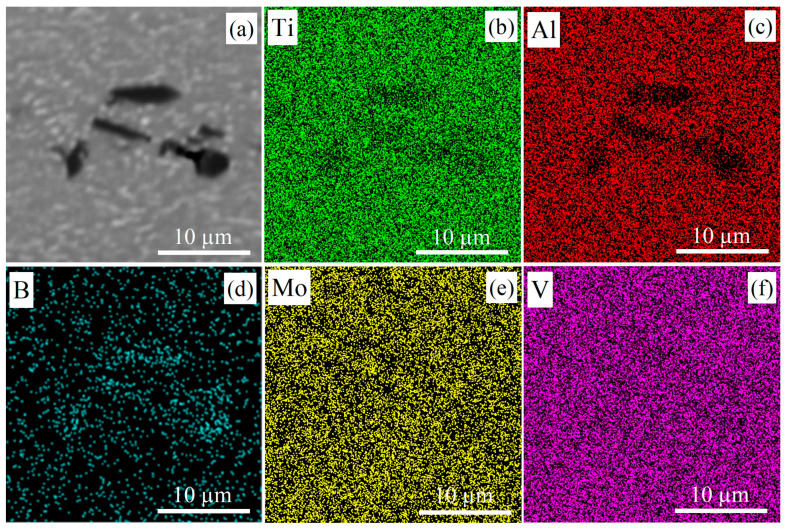
(**a**) SEM images and (**b**–**f**) EDS SEM maps of the hot-rolled 0.1B alloy.

**Figure 5 materials-16-03714-f005:**
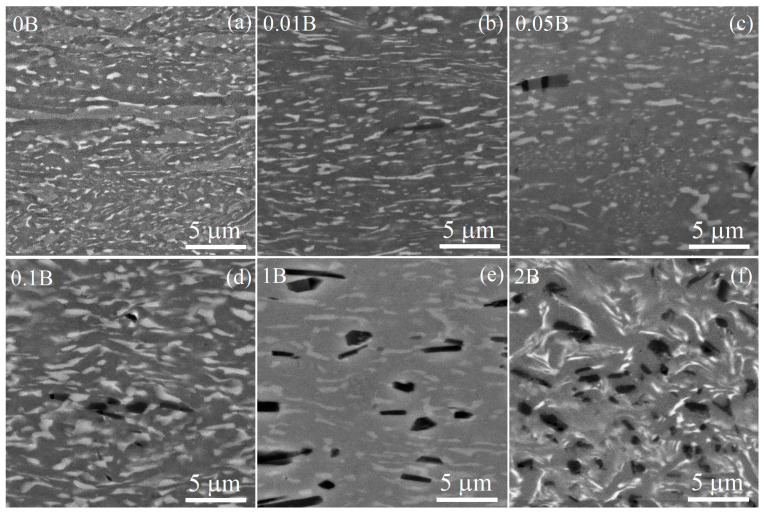
SEM images of the hot-rolled (**a**) 0B, (**b**) 0.01B, (**c**) 0.05B, (**d**) 0.1B, (**e**) 1B, and (**f**) 2B alloys.

**Figure 6 materials-16-03714-f006:**
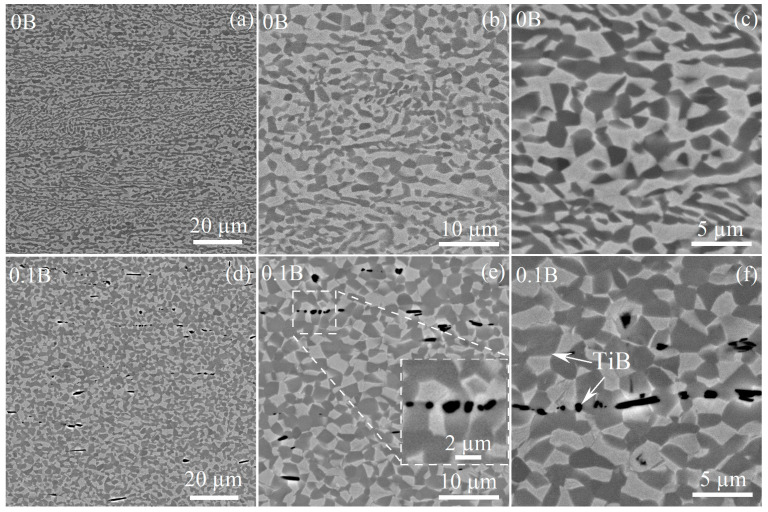
Microstructures of the (**a**–**c**) 0B and (**d**–**f**) 0.1B alloys after annealing for 30 min at 875 °C.

**Figure 7 materials-16-03714-f007:**
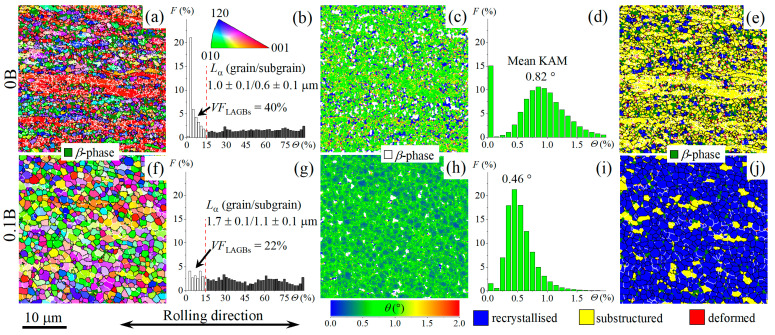
(**a**,**f**) EBSD-IPF maps, (**b**,**g**) grain boundary misorientation angle distributions, (**c**,**h**) kernel average misorientation (KAM) maps, (**d**,**i**) local misorientation distributions and (**e**,**j**) recrystallized fraction map of the (**a**–**e**) 0B and (**f**–**j**) 0.1B alloys after annealing for 30 min at 875 °C.

**Figure 8 materials-16-03714-f008:**
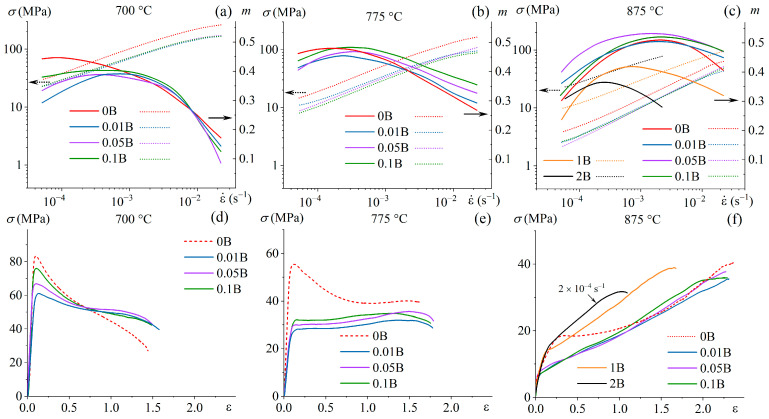
(**a**–**c**) Stress vs. strain rate (dotted lines) and *m* value vs. strain rate (solid lines) obtained by a step-by-step decrease in the strain rate and (**d**–**f**) stress–strain curves at a constant strain rate of 1 × 10^−3^ s^−1^ at (**a**,**d**) 700 °C, (**b**,**e**) 775 °C, and (**c**,**f**) 875 °C.

**Figure 9 materials-16-03714-f009:**
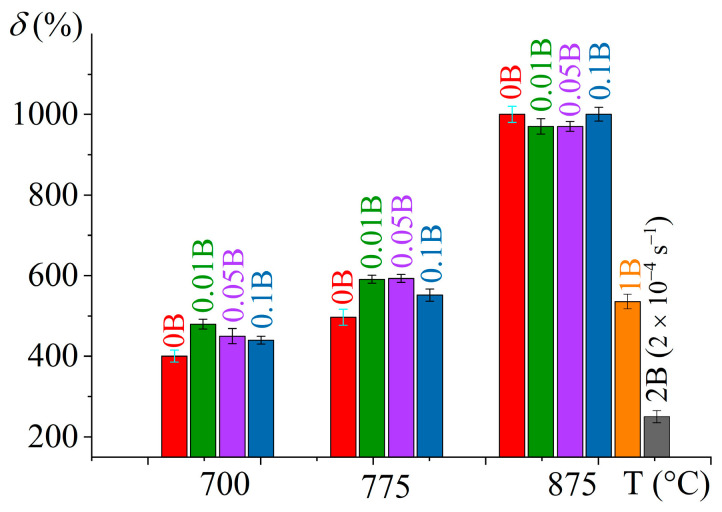
Elongation to failure (*δ*) dependency in a temperature range of 700–875 °C of the investigated alloys.

**Table 1 materials-16-03714-t001:** Chemical composition of the investigated alloys (wt.%).

Alloy	Al	Mo	V	B	Ti
0B	4.2	3.0	1.1	–	Bal.
0.01B	4.1	3.2	1.0	0.01	Bal.
0.05B	4.1	2.9	0.9	0.05	Bal.
0.1B	3.9	3.1	1.1	0.1	Bal.
1B	4.2	3.0	1.0	1	Bal.
2B	4.3	3.1	1.1	2	Bal.

**Table 2 materials-16-03714-t002:** Microstructure parameters of the investigated alloys after solidification.

Alloy	TiB Size (µm)	TiB Volume Fraction *f* (%)	Grain Size (µm)
0B	–	–	700 ± 70
0.01B	0.7 ± 0.2	0.1	490 ± 40
0.05B	1.4 ± 0.3	0.2	410 ± 30
0.1B	2.5 ± 0.4	1.1	210 ± 20
1B	2.5 ± 0.5	5.0	670 ± 50
2B	2.9 ± 0.5	10.8	750 ± 80

**Table 3 materials-16-03714-t003:** Microstructure parameters of the investigated alloys after thermomechanical processing.

Alloy	TiB Size (µm)		*L*_⊥_/*L*_∥_	Grain Size (µm)		Volume Fraction (%)		
	*L* _∥_	*L* _⊥_		*α*	*β*	*α*	*β*	TiB
0B ^*^	–	–	–	1.0 ± 0.2	0.5 ± 0.1	80	20	–
0.01B ^*^	2.1 ± 0.3	0.6 ± 0.1	0.3	1.0 ± 0.1	0.7 ± 0.1	82	18	0.2
0.05B ^*^	2.2 ± 0.4	0.7 ± 0.1	0.3	1.0 ± 0.1	0.7 ± 0.1	80	20	0.3
0.1B ^*^	2.8 ± 0.2	0.7 ± 0.3	0.2	0.9 ± 0.1	0.6 ± 0.1	78	18	1.1
1B ^**^	3.1 ± 0.9	0.7 ± 0.2	0.2	1.2 ± 0.1	0.8 ± 0.1	76	19	5.2
2B ^**^	1.0 ± 0.2	1.0 ± 0.1	1.0	1.0 ± 0.1	0.6 ± 0.1	70	20	10.5

^*^–hot-rolled at 750 °C, ^**^–hot-rolled at 900 °C.

**Table 4 materials-16-03714-t004:** Room-temperature mechanical properties of the investigated alloys after superplastic deformation.

Alloy	*YS* (MPa)	UTS (MPa)	*δ* (%)
100 pct strain at 875 °C/1 × 10^−3^ s^−1^ and air cooling			
0B	770 ± 10	860 ± 10	7 ± 1
0.01B	733 ± 7	840 ± 10	7 ± 1
0.05B	678 ± 8	840 ± 10	7 ± 1
0.1B	680 ± 7	870 ± 5	7 ± 1
1B	830 ± 8	1020 ± 6	1.1 ± 0.4
2B	–	778 ± 5	–
100 pct strain at 875 °C/1 × 10^−3^ s^−1^, water cooling and ageing at 480 °C for 16 h			
0.01B	865 ± 7	982 ± 8	7 ± 1
0.1B	780 ± 7	960 ± 5	5 ± 1

## Data Availability

Not applicable.

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
