# Peer review of "Effect of Boron on the Microstructure, Superplastic Behavior, and Mechanical Properties of Ti-4Al-3Mo-1V Alloy"

_materials, 2023, doi:10.3390/ma16103714_

Round 1

Reviewer 1 Report

This article mainly introduces the influence of 0.01-2%B on the microstructure, superplasticity, and mechanical properties at room temperature of the Ti-4%Al-3%Mo-1%V alloy. The experimental scheme is reasonable and the experimental data can prove the correctness of the conclusion. It is a paper worthy of publication.

Existing problems and the suggestions for improvement:

1. According to lines 24-25 of the abstract, "Heat treatment increased post-forming strength characteristics for the alloys with 0.01- 0.1wt.% boron by 90–140 MPa without ductility reduction". In fact, the addition of 0.01- 0.1wt.% boron effected the plasticity slightly according to Table 4. In addition, the data in Table 8 only showed the samples with the addition of 0.01wt.% B and 0.1wt.%B, so it is not accurate enough to use 0.01-0.1wt.% to describe it.

2. The temperature in figure name is inconsistent with the mark in Fig.8, and the meaning of the solid and dotted lines in Fig. 8(a)-(c) is not specified; In Figure 9, the data shows a range of 700-875℃, which is inconsistent with the 775-875℃ in the title of picture.

3. According to line 297-300 of Chapter 4, "With an increase in boron content from 0.1 to 1-2% the fraction of TiB particles significantly increased and, considering the similar mean size of the boride particles, the Zener pinning effect  during cooling in a β-phase field should be stronger and finer grains should be formed." But according to Table 3, when boron content increases from 0.1 to 1-2%, The average size of β phase increases firstly and then decrease, and no finer grain is formed at the amount of 1wt.% B.

Author Response

Response to Reviewer

This article mainly introduces the influence of 0.01-2%B on the microstructure, superplasticity, and mechanical properties at room temperature of the Ti-4%Al-3%Mo-1%V alloy. The experimental scheme is reasonable and the experimental data can prove the correctness of the conclusion. It is a paper worthy of publication.

We thank the Reviewer for their insightful comments on our manuscript. We have taken all the comments into consideration, and we have marked changes in the manuscript with blue-colored text and the changes are presented step-by-step in this detailed response.

Existing problems and the suggestions for improvement:

Q1. According to lines 24-25 of the abstract, "Heat treatment increased post-forming strength characteristics for the alloys with 0.01- 0.1wt.% boron by 90–140 MPa without ductility reduction". In fact, the addition of 0.01- 0.1wt.% boron effected the plasticity slightly according to Table 4. In addition, the data in Table 8 only showed the samples with the addition of 0.01wt.% B and 0.1wt.%B, so it is not accurate enough to use 0.01-0.1wt.% to describe it.

A.1 According to Table 4 post-forming heat treatment slightly decreased ductility from 7 ± 1 to 5 ± 1% for 0.1B. We have corrected the manuscript as follows:

“Post-forming heat treatment including quenching and ageing increased strength characteristics of the alloys with 0.01 and 0.1 % boron by 90–140 MPa and insignificantly decreased ductility.”

“Quenching and aging of the alloys with a low boron content of 0.01 and 0.1% increased yield strength by 100-130 MPa and tensile strength by 90-140 MPa. For the 0.01B alloy, elongation at fracture was the same after both air cooling and heat treatment but for the 0.1B alloy, ductility slightly reduced after heat treatment from 7 to 5%.”

Q2. The temperature in figure name is inconsistent with the mark in Fig.8, and the meaning of the solid and dotted lines in Fig. 8(a)-(c) is not specified; In Figure 9, the data shows a range of 700-875℃, which is inconsistent with the 775-875℃ in the title of picture.

A2. The meaning of the solid and dotted lines was added in title of Figure 8. Also, the temperature in the name of Figure 9 was corrected to the correct value of 775°C:

“Figure 8. (a-c) Stress vs. strain rate (dotted lines) and m value vs. strain rate (solid lines) obtained by a step-by-step decrease in the strain rate and (d–f) stress–strain curves at a constant strain rate of 1×10−3 s−1 at (a,d) 700 °C, (b,e) 775 °C and (c,f) 875 °C.”

The temperature range (775-875°C) in the caption of Figure 9 was replaced with the correct values of 700-875°C.

Q3. According to line 297-300 of Chapter 4, "With an increase in boron content from 0.1 to 1-2% the fraction of TiB particles significantly increased and, considering the similar mean size of the boride particles, the Zener pinning effect during cooling in a β-phase field should be stronger and finer grains should be formed." But according to Table 3, when boron content increases from 0.1 to 1-2%, The average size of β phase increases firstly and then decrease, and no finer grain is formed at the amount of 1wt.% B.

A3. This discussion considers prior β grain refinement effect in as cast state (Table 2). The alloys with 0.1 and 1%B had the same size of TiB particles but 5 times higher fraction of TiB and 3 times coarse grains are observed for 1%B. Following Reviewer recommendation and to avoid misunderstanding, we have corrected the text as follows.  

“With an increase in boron content the fraction of TiB particles significantly increased (Table 2) and, considering the same mean sizes of the boride particles for 0.1% B and 1% B of 2.5 µm. Zener pinning effect increases with an increase particles fraction and with a decrease particle size [25]. Thus, during cooling in a β-phase field Zener pinning force should be stronger and finer grains are expected. Meanwhile, grains were coarser at a high borides fraction. Thus, the refinement effect could not be explained by Zener pinning mechanism.”

We also referenced Table 2 in the text for clarity:

“A trace boron addition, up to 0.1%, provided an effective refinement of the prior β grains. A mean grain size decreased more than three times (Table 2)…”

Reviewer 2 Report

The manuscript mainly studies the influence of 0.01-2% B(boron) on the microstructure, superplasticity, and mechanical properties at room temperature of the Ti-4Al-3Mo-1V alloy. The manuscript is well presented. However, this work needs to be further modified according to the comments and suggestion as below.

1.       Please check out the entire manuscript, and unify the terms “Ti-4Al-3Mo-1V” and “Ti-4Al-1V-3Mo”.

2.       Please further improve the English writing throughout the article.

3.       It is proposed to indicate the region of hypoeutectic concentration for the studied boron content of 0.01–1% in Figure 1. (a).

4.       To improve the expression ability of the article, it is worthwhile to illustrate the meaning represented by the solid and dashed lines in Figure 8. (a-c).

5.       In Figure 9, the description of 2%B at 700 °C and 775 °C is lacking. Please provide additional descriptions.

Nil

Author Response

Response to Reviewer

The manuscript mainly studies the influence of 0.01-2% B(boron) on the microstructure, superplasticity, and mechanical properties at room temperature of the Ti-4Al-3Mo-1V alloy. The manuscript is well presented. However, this work needs to be further modified according to the comments and suggestion as below.

We thank the Reviewer for their insightful comments on our manuscript. We have taken all the comments into consideration, and we have marked changes in the manuscript with blue-colored text and the changes are presented step-by-step in this detailed response.

Q1. Please check out the entire manuscript, and unify the terms “Ti-4Al-3Mo-1V” and “Ti-4Al-1V-3Mo”.

A1. The terms were unified to Ti-4Al-3Mo-1V or Ti-Al-Mo-V based alloy.

Q2. Please further improve the English writing throughout the article.

A2. We carefully checked the English writing and re-read text to improve the language of the manuscript.

Q3. It is proposed to indicate the region of hypoeutectic concentration for the studied boron content of 0.01–1% in Figure 1. (a).

A3. The Figure 1a was modified and the studied alloys were marked up according to recommendation.

Q4. To improve the expression ability of the article, it is worthwhile to illustrate the meaning represented by the solid and dashed lines in Figure 8. (a-c).

A4. The meaning of the solid and dotted lines was represented by arrows on the Figure 8a-c.

The caption was also corrected as follows:

“Figure 8. (a-c) Stress vs. strain rate (dotted lines) and m value vs. strain rate (solid lines) obtained by a step-by-step decrease in the strain rate and (d–f) stress–strain curves at a constant strain rate of 1×10−3 s−1 at (a,d) 700 °C, (b,e) 775 °C and (c,f) 875 °C.”

Q5. In Figure 9, the description of 2%B at 700 °C and 775 °C is lacking. Please provide additional descriptions.

A5. The 2%B alloy demonstrated non-superplastic behavior at elevated temperatures of 875 °C with a low m value (0.35) and even at a low strain rate of 2×10-4 s-1 (Figure 8f) the elongation to failure was 250%. In this regard, the testing of the 2%B alloy at 700 °C and 775 °C was not reasonable.

Reviewer 3 Report

Dear Authors: "Effect of Boron on the Microstructure, Superplastic Behavior, 2 and Mechanical Properties of Ti-4Al-3Mo-1V Alloy". I found the article interesting and well-prepared scientifically. Comments on the editorial page of the work, which in my opinion should be corrected and supplemented:

A.        2.1 Material Processing

-             There is no reference to Table 1 in the text.

B.         2.3. Tensile tests

-             Please explain the wording:The mechanical properties of the alloys at room temperature were determined on a Zwick Z250 testing machine after 100% deformation at a temperature of 875 °C with a constant strain rate of 1 × 10−3 s−1 on samples with a gauge section size of 6×0.8×25 mm3.”

I propose to correct this sentence at work

C.         3.1. Phase composition analysis

-             In Fig. 2, 200mm and 500mm are plotted on 2 photos. Please complete the rest of the photos. Similarly Fig. 3, 4 and 5. -             Table 2 shows the TiB size. Please specify how the measurement deviation was determined, e.g. (+/-0.2mm). D.        Conclusions -             The conclusions contained in the work, in her opinion, constitute an analysis of the results. This point should include observations and statements about what has been done in the work. Has the assumed goal of the work been achieved, which in my opinion should be indicated.

Author Response

Response to Reviewer

Dear Authors: "Effect of Boron on the Microstructure, Superplastic Behavior, 2 and Mechanical Properties of Ti-4Al-3Mo-1V Alloy". I found the article interesting and well-prepared scientifically.

We thank the Reviewer for their insightful comments on our manuscript. We have taken all the comments into consideration, and we have marked changes in the manuscript with blue-colored text and the changes are presented step-by-step in this detailed response.

Comments on the editorial page of the work, which in my opinion should be corrected and supplemented:

A. 2.1 Material Processing

- There is no reference to Table 1 in the text.

A1. The reference to Table 1 is in the first sentence of first paragraph in 2.1 Material Processing:  The boron-free reference Ti-Al-Mo-V alloy and five alloys with boron additions in a range of 0.01-2 wt.% were prepared (Table 1).”

B. 2.3. Tensile tests

- Please explain the wording: “The mechanical properties of the alloys at room temperature were determined on a Zwick Z250 testing machine after 100% deformation at a temperature of 875 °C with a constant strain rate of 1 × 10−3 s−1 on samples with a gauge section size of 6×0.8×25 mm3.” 

I propose to correct this sentence at work.

A2. The text was corrected as follows:

“The mechanical properties of the alloys at room temperature were determined on a Zwick Z250 testing machine using samples with a gauge section size of 6×0.8×25 mm3. Before cutting the samples, the sheets were pre-strained for 100% at a temperature of 875 °C with a constant strain rate of 1×10−3 s−1. Then two types of treatment were processed; (1) air-cooling from the deformation temperature, and (2) water-quenching from the deformation temperature and further ageing at 480 °C for 16 h. Before testing, the samples were treated in Kroll's reagent for ~20 min for α-case removal.”

C. 3.1. Phase composition analysis

- In Fig. 2, 200mm and 500mm are plotted on 2 photos. Please complete the rest of the photos. Similarly Fig. 3, 4 and 5. - Table 2 shows the TiB size. Please specify how the measurement deviation was determined, e.g. (+/-0.2mm).

A3. Following Reviewer recommendation, scales were added to all microstructures in Figures 2-5. The additional magnification (200 µm) was added on the Fig.2d for identification of the microstructure due to a small grain size for the 0.1B alloy.

The determination of measurement deviations was added to the "Materials and Methods" part:

“The mean size and volume fraction of grains and TiB particles were determined by random linear secants method by averaging more than 200 measurements. The error bars were calculated using standard deviation of a mean value and a confidence probability of 0.95.”

D. Conclusions - The conclusions contained in the work, in her opinion, constitute an analysis of the results. This point should include observations and statements about what has been done in the work. Has the assumed goal of the work been achieved, which in my opinion should be indicated.

A4. The conclusions were corrected in accordance with the recommendation as follows:

“The influence of 0.01-2 wt.% B on the microstructure, superplasticity, and mechanical properties at room temperature of the Ti-4%Al-3%Mo-1%V alloy was studied. The boron effect on the microstructure and properties of the alloys was strongly different for the ranges of 0.01-0.1% B and 1-2% B. The main conclusions are summarized as follows.

The TiB particles of eutectic origin were formed after solidification in the studied B range. In as-cast alloys, the mean size of borides increased from 0.7 to 2.5 µm with an increase B from 0.01 to 0.1% and insignificantly changed in a range of 0.1-2% B. The volume fraction of the TiB phase increased from ~0.1% at 0.01% B to ~11% at 2% B that agreed to Thermo Calc simulation. Trace boron addition significantly reduced the mean size of prior β grains from ~700 µm forthe B-free alloy to ~210 µm for the alloy with 0.1% B. Grain refinement was not revealed and a mean grain size was ~670-750 µm for the alloys with 1 and 2% B.

After thermomechanical treatment with a final hot rolling, the alloys exhibited similar elongated grains of a mean size in a range of 0.9-1.2 µm for the a phase and 0.5-0.8 µm for the β phase. For the B-free and minor B alloys with TiB volume fraction below ~1.1%, the strain rate sensitivity coefficient m was ~0.5, and the elongation to failure was ~500-1000% at temperatures of 775-875 °C and a strain rate of 1×10-3 s-1 but B influenced deformation behavior. Due to the facilitation of recrystallization and globularization effects, the 0.01-0.1B decreased flow stress values at the initial stage of superplastic deformation and provided a stable flow. The effect was significant at a low deformation temperature of 775 °C. Along with this, owing to the stimulation recrystallization effect, a trace B addition decreases post-forming strength properties and insignificantly influenced ductility at room temperature; yield strength decreased from 770 MPa for the B-free alloy to 733 MPa for 0.01% B, and to 680 MPa for 0.1B. Quenching and aging increased the strength properties by 90-140 MPa and slightly decreased ductility for 0.1B. The modifying of Ti-Al-Mo-V alloy with 0.01B provided both uniform and stable flow during superplastic deformation and good mechanical properties at room temperature.

The alloy with 1% B and ~5.2% of TiB whiskers of a mean thickness ~0.7 µm and a mean length of ~3 µm exhibited superplasticity at elevated deformation temperature of 875 °C with m of 0.5 and elongation to failure of ~500% and room temperature yield strength of 830 MPa, ultimate strength of 1020 MPa but low elongation at fracture of 1.1%. An increase boron content to the eutectic concentration of 2 wt.% increased a TiB particles fraction to ~11% and provided the particles with a mean size of ~1 μm after thermomechanical processing. The high-boron eutectic alloy demonstrated a non-superplastic behavior at elevated temperatures and a brittle fracture at room temperature with a strong decrease in the ultimate tensile strength to 780 MPa.”

Round 2

Reviewer 1 Report

In this paper, the role of element B is discussed. The experimental data are detailed and reasonable. It can provide a good reference for the workers studying titanium alloy and is worth recommending for publication.

Reviewer 2 Report

The authors have revised the manuscript according to the comments raised.

Reviewer 3 Report

Dear Authors. Thank you for making corrections to the article. Now the work in my opinion is correct.